# LEARNING PREDICTIVE, ONLINE APPROXIMATIONS OF EXPLANATORY, OFFLINE ALGORITHMS

## ABSTRACT

In this work, we introduce a general methodology for approximating offline algorithms in online settings. By encoding the behavior of offline algorithms in graphs, we train a multi-task learning model to simultaneously detect behavioral structures which have already occurred and predict those that may come next. We demonstrate the methodology on both synthetic data and historical stock market data, where the contrast between explanation and prediction is particularly stark. Taken together, our work represents the first general and end-to-end differentiable approach for generating online approximations of offline algorithms.[1]

## 1 INTRODUCTION

As the saying goes, *hindsight is 20/20*. With symmetric access to past and future data, it's easier to see paths joining cause and effect. Exploiting this symmetry, offline algorithms can recognize patterns only visible in retrospect and produce rich explanations of historical data. However, those same algorithms fall short in real-time settings, where future data is inaccessible. To bring an offline algorithm *online*, it must either be manually adapted or its requirement for future data satisfied through exogenous forecasting techniques. To more efficiently bridge this gap between offline algorithms and their online counterparts, we introduce an end-to-end differentiable methodology for approximating offline algorithms in real-time.

While a variety of approaches exist for translating *particular* offline algorithms into their online counterparts, ours is the first to approach the problem in the general case (Blum, 1994; Kakade & Kalai, 2006; Ben-David et al., 1997). Using a graph-structured labeling scheme, our method blends explanation and prediction to approximate the offline requirement for future data. However, unlike most time series forecasting techniques, which predict trajectories of time series data directly, we instead focus on the *behavior* that an offline algorithm may yield on that data (Mahalakshmi et al., 2016). Refocusing in this way is productive for several reasons: 1) it simplifies the prediction task, which allows us to forecast further into the future 2) it predicts the behavior of an algorithm explicitly designed to ignore irrelevant noise, and 3) it recasts the offline-to-online transformation as a straightforward mapping problem in which we can exploit the end-to-end differentiability of modern machine learning (ML) tools to find a solution.

In our framework, any modeling scheme in which future data informs the representation or interpretation of the past is amenable to approximation. Under this definition, the scope of offline analysis extends well beyond computer science and into fields from medicine to economics.

Medical diagnostics is likely the largest suite of offline algorithms in use today. As patients experience new symptoms or additional test results arrive, both diagnosis and prognosis may be updated in response - meaning symptoms are ascribed to their cause retroactively. Processes for making these updates are codified in diagnostic manuals and revised as new scientific and patient data become available (Jeffery et al., 2012). However, with rapidly growing volumes of clinical data, it is becoming increasingly difficult for individual healthcare providers to integrate the new information, and with clinical outcomes closely tied to the accuracy of these offline diagnostic procedures, finding ways to improve them is of interest to both the medical community and the public at large (Berger, 1999; Balogh et al., 2015).

---

[1]All code, data, and configuration scripts are available under the MIT License at [GitHub repository redacted for review].

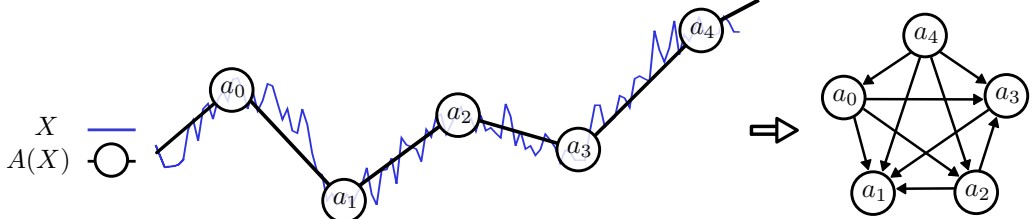

Figure 1: Left: 1D time series $X$ and the output of an offline piece-wise linear approximation (PLA) algorithm $A(X)$. Right: The graphical decision-point representation (Section 2.1) of the structure $\{a_0, \cdots, a_4\}$ using a *greater-than* relation between decision points.

Similarly, when considering economic and societal health, economists and policy-makers have long relied on offline algorithms to interpret historical data. Recessions and credit crises appear obvious in hindsight, but as many of these offline techniques lack online counterparts, forecasting similar events in real-time is difficult (Colander, 2010). Indeed, economic models are often criticized for their overreliance on explanatory processes when making forecasts, which may in part be driven by the absence of a general framework for converting explanatory models into predictive ones (Lipsey, 2001). With these forecasts driving trillions in government and private spending annually, consequences of inaccuracy are wide-ranging, and everyone stands to benefit from more informed, data-driven predictions (Lewis & Pain, 2015). In this paper, we'll take a step towards these data-driven, *predictive* economic models by using our methodology to forecast behavioral structures in equity markets.

We begin Section 2 by formalizing the notion of *behavioral structures* in characterizing offline algorithms. We explain how to use these behavioral structures to generate descriptive labels and then formulate the machine learning problem in those terms. In Section 3, we review the offline algorithm being approximated in our experiments on synthetic and real-world data. We then detail the multi-task ML model used to demonstrate the methodology. We note that the experiments are designed to build intuition for the methodology, and as such are confined to 1D settings. However, in Section 4.1 we survey additional examples of single and multidimensional algorithms and explain how our method can be equally applied to any offline algorithm on time series data. We conclude by reviewing Related Work, discussing directions for future research, and outlining the steps we've taken to ensure reproducibility. Supplementary model architecture and training details are available in Appendix A.1.

## 2    GENERAL SCHEMA

Here, we present the building blocks of our method and develop intuition with illustrative examples motivated by the experiments in Section 3.

### 2.1    GRAPHICAL DECISION-POINT REPRESENTATION

Suppose we have a time series $X = \{x_1, \ldots, x_T\}$, where $x_i \in \mathbb{R}$, and an offline algorithm $A(\cdot)$ which operates on $X$ to produce a sequence of decision points $A(X) = \{(x_j, a_{x_j}), \cdots, (x_k, a_{x_k})\}$, as in Figure 1. Note that $A(X)$ can yield decision points $a_{x_j}$ on only a subset of $x_j \in X$. We define a behavioral structure $s$ to be an ordered set of $n$ contiguous decision points in $A(X)$:

$$s = \{(x_{k-n+1}, a_{x_{k-n+1}}), \ldots, (x_k, a_{x_k})\} \tag{1}$$

where $k$ is the position of the terminal decision point. These structures are what the ML model will ultimately learn to classify, and as such we need a representational scheme to meaningfully differentiate structures $s \in S$ (the set of all unique structures in the dataset). To do so, we represent each structure $s$ as a directed graph in which each decision point is a vertex and edges are defined by a *relationship-indicator* function $r$. This indicator function will be application-dependent and can take any form that meaningfully differentiates behavioral structures in $S$. Once $S$ has been populated with all the structures yielded by $A(X)$, each unique structure $s_z \in S, |S| = k$, is assigned a class label $C_z$.

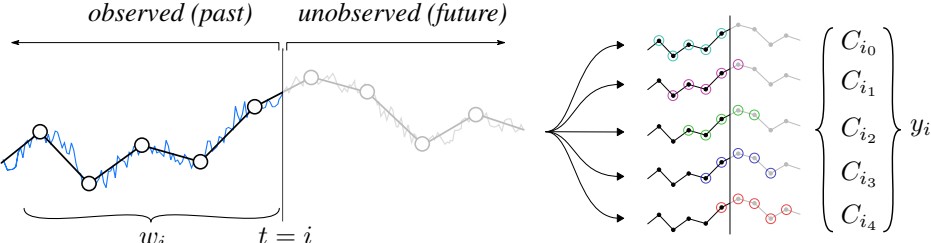

Figure 2: Left: Extending the example from Figure 1, we show $n = 5$ decision points visible in $w_i$. Decision points in the future (at times $t > i$) are not visible. Right: The decision point structures of the next $n$ structures beginning in $w_i$, followed by their corresponding class labels in $C_{i_\lambda} \in y_i$. Highlighted decision points define the progressively forward-looking structures we aim to classify.

The offline algorithm $A(\cdot)$ in Figure 1 is a piece-wise linear approximation (PLA) algorithm, applied to a 1D time series $X$. To construct the graphical decision-point representation of the $n = 5$ visible decision points, we choose the *relationship-indicator* function $r$ to be the "greater than" relationship[2]. Directed edges will then extend from nodes in $s$ with higher values to those with lower values.

In the next section, we explain how to use this labeling scheme to characterize structures extending from the observed past into the unobserved future.

## 2.2 PROGRESSIVELY FORWARD-LOOKING TASKS

For each $x_i \in X$, we define $w_i = \{x_{i-d+1}, \dots, x_i\}$ to be the window on $X$ of size $d$ terminating at $x_i$. As shown in Figure 2 (Left), this window contains $n$ decision points, or one full structure. Increasingly forward-looking structures are those which begin in $w_i$ and end in the unobserved future $t > i$. Figure 2 (Right) highlights increasingly forward-looking structures which we refer to by the number of decision points $\lambda$ that lie in the unobserved future. The structure entirely visible in $w_i$ becomes $C_{i_0}$, while the structure with four unobserved decision points is $C_{i_4}$. Each window $w_i$ is labeled with all the structures that begin within it $y_i = \{C_{i_0}, \cdots, C_{i_{n-1}}\}$. The class label assigned to each $C_{i_\lambda} \in y_i$ represents one of $k$ unique structures found in the original dataset.

From here, we refer to the task of labeling the $\lambda^{th}$ structure on $w_i$, i.e. generating a prediction $\hat{C}_{i_\lambda}$, as Task-$\lambda$. For each $w_i$, our goal is to predict these increasingly forward-looking structures $\{C_{i_0}, \cdots, C_{i_{n-1}}\} \in y_i$.

## 2.3 PROBLEM FORMULATION

Suppose again that we have a time series $X = \{x_1, \dots, x_T\}$, where $x_i \in \mathbb{R}$, and an offline algorithm $A(\cdot)$ which operates on $X$ to produce a sequence of decision points $A(X) = \{(x_i, a_{x_i}), \cdots, (x_j, a_{x_j})\}$, as in Figure 1. Further suppose we have identified and labeled each of the $k$ unique structure classes present in the dataset, as described in Section 2.1.

For each window $w_i$ our goal is to find an estimator which yields class conditional probabilities over all $k$ classes for each Task-$\lambda$. Formally, we are looking for an estimator $f$:

$$f : w_i \in \mathbb{R}^d \rightarrow \{C_0, \dots, C_{k-1}\}^\lambda, \text{ where } f(w_i) = y_i. \tag{2}$$

This formulation closely resembles the multi-task learning problem, as generating predictions for each of the $\lambda$ structures represent related but fundamentally distinct tasks. Task-0 is entirely explanatory, as all relevant decision points are visible in $w_i$, while Task-(n-1) is entirely predictive, as only a single decision point is visible.

---

[2]In this example $r$ is a transitive relationship, and the graph can be simplified into an ordered list (smallest to greatest) but we note that this labeling scheme can generalize to graphs of arbitrary size and relations of arbitrary dimension.

In the following section, we outline experiments in which our methodology is used approximate a common offline time series algorithm using a hard parameter sharing multi-task ML model.

## 3 EXPERIMENTS

Here we detail the offline algorithm we aim to approximate, as well as the synthetic and real-world datasets we will use to train our model. Experiments on synthetic data are designed to verify the functionality of our method in a controlled setting, and experiments on historical market data show the method's utility on noisy real-world data.

To demonstrate the method, we use a simple and well-understood multi-task learning model inspired by the hard parameter sharing approach introduced by Caruana (1993). While this architecture performs well in experiments on both synthetic and real-world data, we choose it primarily because it demonstrates the role of machine learning in our framework: to map behavioral structures in the past to those yet to come. The model is a multi-headed, multi-task learning model, with each of the $\lambda$ heads performing a single Task-$\lambda$. Full model details are available in Section 3.3.

### 3.1 OFFLINE ALGORITHM: $\ell_1$ TREND FILTERING

$\ell_1$ trend filtering ($\ell_1$TF) is a batch piece-wise linear approximation (PLA) method (Kim et al., 2009) which extracts linear components from a time series by minimizing the following objective function:

$$\frac{1}{2}\sum_{t=1}^{T}(x_t - \hat{x}_t)^2 + \rho\sum_{t=2}^{T-1}|\hat{x}_{t-1} - 2\hat{x}_t + \hat{x}_{t+1}|, \tag{3}$$

where $x_t$ indicates a length-$T$ time series, $\hat{x}_t$ denotes the approximation, and $\rho$ is a smoothness parameter. In equation 3, as the $\ell_1$-norm is used for smoothness, the resulting approximation is piece-wise linear which makes it suitable for structure and change point detection (CPD) (Aminikhanghahi & Cook, 2017). Decision points are then defined by the intersection of neighboring segments produced by equation 3. Crucially, this algorithm requires access to future states, and as such is strictly confined to offline settings.

In our experiments, decision points $a_{x_j} \in \mathbb{R}$ are generated by the $\ell_1$TF algorithm and correspond to points in the time series at which the slope changes significantly. The value of each decision point is assigned to the value of the time series at that position, as shown in Figure 1.

We choose the $\ell_1$TF algorithm for it's clarity and utility across domains. However, in Section 4.1 we review additional algorithms and explain how the method may be applied in multi-dimensional settings.

### 3.2 DATASETS

To validate our methodology, we use synthetic and real-world datasets. In both experiments, structures have $n = 5$ decision points, yielding $\lambda = 5$ tasks as shown in Figure 2.

#### 3.2.1 SYNTHETIC DATA

Our synthetic dataset $X_{Synth} \in \mathbb{R}^1$ is 1D time series as shown in Figure 1. Structural sequences in $X_{Synth}$ are composed of $n = 5$ decision points with a window size $d = 150$. As these are generated algorithmically, we can produce structural sequences with precise amounts of randomness $\gamma$ in their ordering. When $\gamma = 0$, the structure sequence is deterministic, meaning that any structure class $C_{i_0}$ is always followed by the same sequence of structures $C_{i_1}, \cdots, C_{i_{n-1}}$. When $\gamma = 1$, there are no meaningful temporal relationships between structures and the trajectory of $X_{Synth}$ is entirely random. Once all the structures (10k in total) are generated, we normalize $X_{Synth}$ to have mean zero and standard deviation 0.1. Then we jitter the series with Gaussian noise having standard deviation 0.02, and split it into non-overlapping[3] train/validation/test sets in percentages 85%/1%/14% respectively.

---

[3]The non-overlapping requirement is crucial, as overlap between windows in the training/validation/testing sets could skew the final test results.

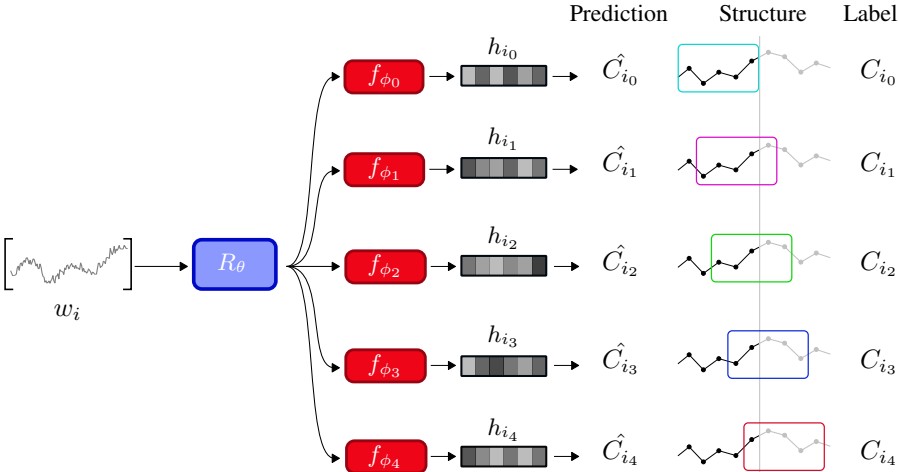

Figure 3: The hard parameter sharing multi-task network for the case in which the structure size $n = 5$, where $n$ is both the number of decision points within a structure and the number of heads of the network. Under "Structure" we show the target structures for each Task-$\lambda$.

We construct the test set such that it contains only structures present in the train set (so we can recognize them in Task-0), but with their *sequential relationships* randomized by the factor $\gamma$.

Decision points are produced using the $\ell_1$TF algorithm and structures are labeled using a *greater-than* relationship indicator function $r$ as in Figure 1. Each label $y_i \in \{C_{i_0}, \cdots, C_{i_{n-1}}\}$ is constructed as per the procedure in Section 2.2, and contains the visible structure's class $C_{i_0}$ as well as the four class labels for structures beginning in $w_i$ and extending into the future.

### 3.2.2 HISTORICAL STOCK MARKET DATA

Our real-world dataset $X_R \in \mathbb{R}^1$ is also a 1D time series as shown in Figure 1. The raw data is price-action (changes in the price of a security) in one minute intervals for five commonly traded stocks on the NYSE, and covers all trading days[4] between 2004 and 2020[5]. Decision points are generated using the $\ell_1$TF algorithm and labels for each structure are assigned according to the procedure outlined in Section 2.2. Once the data is processed and labeled, it is normalized to have mean zero and standard deviation 0.1, then split into non-overlapping train/validation/test sets in percentages 85%/1%/14% respectively. For this real-world data, structures which cover more or less than 150 points are resized to have length 150. In total, we identify 190k structures of size $n = 5$ and 32 unique structure classes in the data.

### 3.3 MODEL ARCHITECTURE

Figure 3 details the multi-task ML model architecture. While this particular multi-task architecture was suitable to demonstrate the method, we emphasize that our method is general, and allows for arbitrary architectures to be implemented depending on the demands of the offline algorithm being approximated.

We adopt a hard parameter sharing scheme as introduced by Caruana (1993) for its simplicity, intrinsic regularization, and its performance in our experiments on synthetic and real world data. The representation learner $R_\theta$ is convolutional neural network (CNN) parameterized by shared parameters $\theta$ that learns a representation of the input sequence $w_i$. It is composed of three max-pooled 1D convolutional layers followed by a batch normalization layer (Ioffe & Szegedy, 2015). Each of the $\lambda$ tasks is performed by an MLP $f_{\phi_\lambda}$ parameterized by task-specific trainable parameters $\phi_\lambda$ and maps the representation learned by the CNN to class conditional probabilities over all $k$ classes. Full

---

[4]As the distribution of price changes is significantly different after trading has closed, only data during market hours are used.

[5]Financial data obtained through Polygon.io.

details for each layer of the representation learner as well as the MLP heads are available in Figure 4 in the Appendix. Compactly, the output of each of the heads of the network can be described as follows:

$$h_{i_\lambda} = f_{\phi_\lambda}\Big(R_\theta(x)\Big) \tag{4}$$

The network described in equation 4 is an implementation of equation 2 that minimizes the Cross-Entropy Loss $L(\cdot)$ between $h_{i_\lambda}$ and the ground truth labels $C_{i_\lambda} \in y_i$ (Zhang & Sabuncu, 2018). Crucially, losses $L(h_{i_\lambda}, y_{i_\lambda})$ are calculated for each task $\lambda$ individually to preserve the option to use different loss functions for different tasks if the application warrants it.

### 3.4 Training Methodology

The model is built in PyTorch and weights are initialized randomly to begin training. Once the datasets are prepared, we train for 300 epochs with batch size = 128 and use the Adam Optimizer with learning rate 0.001 (Paszke et al., 2019; Kingma & Ba, 2015). To account for slight class imbalances, we pass label weights (inverse frequency) to the Cross Entropy Loss criterion. For a complete description of all relevant model and training hyperparameters, see [GitHub repository redacted for review] or Appendix A.1.

## 4 Results and Discussion

Our results demonstrate the method's ability to capture meaningful temporal relationships between behavioral structures identified by offline algorithms, *and* to apply those learnings to make quality predictions in noisy, real-time settings. We present the results in terms of Top-N accuracy for each Task-$\lambda$.[6]

**Synthetic Data:** The goal of the synthetic experiments is to verify the functionality of the method. As expected, increasing the randomness in the sequential relationships between behavioral structures resulted in a proportional decrease in model accuracy on Task-$\lambda$ as $\lambda$ increased. Table 1 shows the steady dropoff in performance as the sequences of behavioral structures become more random.

| | Task-0 | | | Task-1 | | | Task-2 | | | Task-3 | | | Task-4 | | |
|---|---|---|---|---|---|---|---|---|---|---|---|---|---|---|---|
| Top-N Acc | 5 | 2 | 1 | 5 | 2 | 1 | 5 | 2 | 1 | 5 | 2 | 1 | 5 | 2 | 1 |
| $\gamma = 0$ | 100 | 100 | 100 | 100 | 100 | 100 | 100 | 100 | 100 | 100 | 100 | 100 | 100 | 100 | **100** |
| $\gamma = 0.5$ | 72.5 | 72.5 | 72.4 | 72.4 | 72.4 | 72.3 | 72.2 | 66.0 | 64.8 | 72.0 | 59.4 | 56.8 | 71.8 | 53.6 | **50.0** |
| $\gamma = 1$ | 89.9 | 71.5 | 51.0 | 90.4 | 71.1 | 51.2 | 80.1 | 35.1 | 18.4 | 23.2 | 10.0 | 5.7 | 4.7 | 1.6 | **0.7** |

Table 1: Top-N accuracy (shown in percent) for each Task-$\lambda$ on synthetic datasets generated with varying amounts of randomness $\gamma$ in the structure sequences. As expected, the model is able to perfectly predict even the most forward-looking task when the structure sequences are deterministic ($\gamma = 0$). Similarly, when the structure sequences are entirely random ($\gamma = 1$) the model is unable to make any meaningful predictions.

As Task-0 is simply identifying the structure in view, this accuracy stays fairly high as $\gamma \to 1$. The non-monotonicity of the Top-5 accuracies for Tasks 0-2 as $\gamma \to 1$ is driven by two factors: 1) the number of unique structures $|S| = k$ in the dataset increases in proportion to the randomness in structure sequences, and 2) the output size of each head, and thus the total number of trainable parameters in the model, must increase to accommodate the increased number of labels. We review this limitation more thoroughly in Section **??**. The performance of the most forward-looking task, Task-4, falls linearly as $\gamma \to 1$. This is expected, as Task-4 is entirely predictive (4/5 decision points being out of view leaves no explanatory component). Top-1 Accuracies for these entirely predictive results (bolded) verify the method is working properly, and learning *up to* the randomness inherent in the structural sequences.

---

[6]Note: Random initializations across experiments did not appreciably affect the final performance so, for clarity, we have left standard errors out of the main paper. For complete information on the Standard Errors for each Task, see Appendix A.1.4.

| | Task-0 | | | Task-1 | | | Task-2 | | | Task-3 | | | Task-4 | | |
|---|---|---|---|---|---|---|---|---|---|---|---|---|---|---|---|
| | 5 | 2 | 1 | 5 | 2 | 1 | 5 | 2 | 1 | 5 | 2 | 1 | 5 | 2 | 1 |
| Top-N Acc | 98.8 | 91.1 | 71.8 | 94.2 | 70.2 | 45.8 | 79.0 | 39.1 | 18.9 | 45.2 | 16.2 | 7.7 | 30.1 | 12.2 | 6.2 |

Table 2: Top-N accuracy (shown in percent) for each Task-$\lambda$ on the real-world price action data.

**Historical Stock Market Data:** We demonstrate our methodology on equities data for two reasons:
1) the structures in 1D time series (often referred to as "patterns" in this context) are clear and
understandable, and 2) the notion that some market structures are predictive of future behavior is
widely held in the trading community (Bremer et al., 1997; Niarchos & Alexakis, 2003). While this
notion of structural predictability exists, studies of such structures remain confined to a small subset
of easily recognizable structures, and do not approach the problem in the general case (Savin et al.,
2006). As our methodology explicitly learns temporal relationships between *all* structures in the
dataset, it is particularly well suited to testing the original notion that past behavioral structures are
predictive of future behavior. Table 2 shows the Top-N accuracy of our method on the real-world
dataset presented in Section 3.2.2.

Examining the forward-looking tasks, we see that there do exist meaningful relationships between
behavioral structures in equities data and that our model was able to learn them with Top-1 accuracies
well above chance (the dataset contained $k = 32$ unique structures). As structures in raw price-
action data are noisier than those in the synthetic dataset, Top-1 Accuracy on explanatory tasks are
lower while Top-5 accuracies remains high. And while it is beyond the scope of this paper, one
could continue this analysis and investigate prediction performance on a structure-by-structure basis,
comparing the results with the existing literature on the predictive power of price-action patterns.

This experiment also demonstrates how the methodology can be used to express common problems in
the target domain. For example, a popular target when predicting stock prices is to judge the direction
the price of a stock will move at any moment (Shen & Shafiq, 2020). In our framework, this question
could be addressed by comparing the relative values of the decision points predicted by Task-1. If the
$n^{th}$, or terminal, decision point predicted by Task-1 was greater than the $(n-1)^{th}$, or second to last,
decision point in Task-1, then the prediction would be that the price would increase. Similarly, if
the terminal decision point in Task-1 were lower than the second to last, the prediction would be a
price decrease. Investigating the Top-1 predictions of Task-1 in the test set, our model predicts the
direction correctly 99.2% of the time. The difference between this value and the accuracy reported in
Table 2 stems from the similarities between candidate structures. When structures differ by only one
or two decision points, the model can struggle to decide between them.

The directional predictions of Task-1 meet or exceed the performance of many state-of-the-art
ML-based stock price prediction systems (Rezaei et al., 2021). However, the result is made more
interesting when we recognize that our prediction reflects not the usual one-*step*-ahead prediction,
but one-*decision-point*-ahead, which is an average of 16.2 steps ahead in our dataset.

## 4.1 BEYOND THE $\ell_1$ TF ALGORITHM

Offline change point detection (CPD) algorithms are designed to locate transitions between states in
a time series, and are implemented in a wide range of domains from medical condition monitoring
and meteorology to finance and human activity analysis (Truong et al., 2020). While our experi-
ments implement the $\ell_1$ TF algorithm detailed in Section 3.1, our method may be readily applied
to approximate any algorithm that generates such change points, without any a priori knowledge
of the algorithm's inner workings. Below, we list two examples from a large class of offline CPD
algorithms. For an exhaustive survey of CPD algorithms, see van den Burg & Williams (2020).

*Relative Density-Ratio Estimation*: Introduced by Liu et al. (2013), this algorithm solves the CPD
problem using a non-parametric divergence estimation between time series samples from retrospective
segments. The authors demonstrate their methodology on human-activity sensing, speech and Twitter
messages.

*Energy Change Point*: Also introduced in 2013 (Matteson & James, 2013), this algorithm solves
the *multivariate* CPD problem using hierarchical clustering in both divisive and agglomerative

configurations. The authors demonstrate both methods on genetics and financial data, and show performance matching existing state of the art CPD algorithms while making fewer assumptions.

Additionally, our method requires an appropriate relationship indicator function to define behavioral structures. Fortunately, these relations can be trivially obtained from existing knowledge about the target algorithm and data.

*Relationship Indicators*: As change points represent transitions between states, the relationship indicator function must encode meaningful relations between those states. For human activity analysis, relationship indicator functions might be relative miles walked or calories burned (more/less/same), or transitions between "active" and "not active" states (Aminikhanghahi & Cook, 2017). We note that these relationship indicator functions can be applied equally in multivariate settings. For example, in medical condition monitoring, states are routinely defined as a function of multiple variables. One of the many such states is an unusual pattern of breathing known as Kussmaul breathing (Sapra et al., 2021) which is known to indicate underlying pathology and is a function of both the depth of the breath *and* the breathing rate. Using existing multidimensional indicator functions defined in diagnostic procedures, one could easily map change points within each time series (depth of breath and breathing rate) onto behavioral structures and, in combination with the historical change point data, use our method to predict the onset of multivariate states like Kussmaul breathing.

## 5 RELATED WORK

While existing literature does not address the problem of converting offline algorithms to their online counterparts in the general case, a variety of approaches have been proposed for specific algorithms. In this section we review those approaches as well as frameworks introduced to study the differences between explanatory and predictive models.

### 5.1 BATCH TO ONLINE CONVERSION

Blum (1994) showed if one-way predictor functions exist for a given problem, regarding polynomial-time learning, online models are harder to construct than a similar offline model. Kakade & Kalai (2006) proposed an algorithm for converting a *proper*[7] batch algorithm into a transductive online algorithm using *hallucination* to generate random labels for unlabeled data, though only in the agnostic (unrealizable) setting (Ben-David et al., 1997; Kearns et al., 1994; Blum & Hartline, 2005). This algorithm is computationally expensive in practice as it re-runs the batch algorithm for every online prediction. Hutter (2014) proposed two methods, namely Limit and Mixture, converting offline probability estimators into online probability estimators. Limit needs to compute a $\lim_{\rightarrow +\infty}$ which may be incomputable and is not guaranteed to exist. Mixture, which uses Bayesian mixture, provides good performance guarantees (unlike Limit), but they do not provide a way to convert it to an efficient algorithm (Santhanam, 2006).

### 5.2 EXPLANATORY VS. PREDICTIVE MODELS

Shmueli (2010) differentiated predictive models from explanatory models based on four aspects: causasion-association, theory-data, retrospective-prospective, and bias-variance. Sainani (2014) also differentiated these two type of models according to goal, threats to validity, candidate variables, variable selection, measures of model performance, and validation. Sriboonchitta et al. (2019) proved there exists no algorithm generating the best explanatory model for a given data, while there does exist an algorithm generating the best predictive model for a given data. This is why the best predictive models are usually not similar to the best explanatory models.

## 6 CONCLUSION AND FUTURE DIRECTIONS

In this work, we introduced the first end-to-end differentiable methodology for learning predictive models of descriptive algorithms, which is both novel in it's generality and practically useful in many

---

[7] A proper algorithm is defined as an algorithm whose output is always a hypothesis $h \in \mathcal{F}$, where $\mathcal{F}$ is a class of functions.

domains. We described how offline algorithms can be used to generate descriptive training labels and formulated the approximation as a multi-task learning problem. In experiments on synthetic and real-world data, we showed our method is capable of learning *up to* the randomness inherent in the structural sequences, and that it can capture meaningful temporal relationships between behavioral structures even in the noisy price-action of commonly traded equities. Additionally, we showed that the model is capable of matching state-of-the-art performance in one-step-ahead stock price prediction, with the caveat that our method did so using a one-*decision-point*-ahead approach, which translates to 16.2 steps in our dataset. Future research will be directed towards approximating offline algorithms which operate on higher dimensional and potentially multi-modal time series data.

Given the abundance of explanatory, offline algorithms deployed in high-impact settings, there is a noted absence of research into methods for transforming these algorithms into their online counterparts. We hope that this research serves as a stepping stone towards more general inquiry into the problem of translating explanatory processes into predictive ones.

### 6.1 ETHICS

Offline algorithms are used in many contexts and towards many ends, usually to explain and interpret the past. As their scope is so broad, we confine our ethics review to the application of our method in medicine and economic policymaking, where sophisticated diagnostic algorithms are routinely used and where new predictive models could have significant societal impact.

*Medicine:* Electronic health records (EHRs) are sparse, multi-dimensional time series, and the structures present between those series determine what cause is ultimately assigned to each symptom: *"Symptom X appeared, then test result Y arrived, so we know now that X was actually a result of condition Z. Now we know to look for symptom X' as Z progresses.".* This process is precisely what our methodology is well suited for: learning relationships between retroactively defined structures in data and forecasting which structures may come next. In regards to explanation, this methodology could be used to augment existing diagnostic aids and improve patient outcomes by generating better candidate diagnoses more quickly (Elkin et al., 2010). In regards to prediction, this could be integrated into existing prognostic models for predicting disease progression (Vogenberg, 2009). Broadly, by appropriately encoding EHR data, our methodology could help doctors narrow the field of candidate diagnoses and better predict disease progression (Ross et al., 2014).

*Policy:* Economic models are often based on inappropriate assumptions and lack empirical validity (Lipsey, 2001). By encoding historical events and their precedents, e.g. credit crises and interest rates, as multivariate time series, our methodology could serve as a data-driven way to learn the temporal relationships between various economic phenomena and produce better forecasts. Additionally, as the number of predictor variables has exploded in recent years, prediction is becoming more difficult, making an ML-based solution more appealing (Elliott & Timmermann, 2008).

We note that the potential negative impacts of our methodology are the same as those posed by ML at large. Namely, we must take care to understand the distributional qualities of our input data and test appropriately to map out the edge cases. Particularly in healthcare, exploring and understanding edge cases is of paramount importance.

## 7 REPRODUCIBILITY

To ensure reproducibility, we provide thoroughly tested bash and python scripts for every step of the data acquisition, model training, model inference, and experimentation process. We additionally ensure that our experiment code is agnostic of the underlying hardware, and can be run on either CPU or GPU. Our GitHub repository [Redacted for review] contains the following:

1. Detailed README with a clear description of the repo, the methodology, all experiments, and how to interpret new results.
2. Bash scripts to build and configure Conda environments with all required packages.
3. Clear configuration files in which all hyperparameters can be viewed and edited.
4. Full synthetic and equities datasets to reproduce each experiment *plus* scripts to pull additional equities data if desired.

5. Pre-trained models to validate end-to-end functionality.

6. Python scripts for training new models, along with instructions for how additional data should be formatted.

*Note to reviewers:* we have not included processed data in the supplemental material as it would make the file prohibitively large, however, all data is available for download from our GitHub repository once public.

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

# A   APPENDIX

## A.1   TRAINING METHODOLOGY

All code and data are available under the MIT License at [GitHub repository redacted for review]. In this Appendix, we review relevant training hyperparameters and briefly discuss the structure of each of the model components. Hyperparameters were chosen based on a combination of a restricted, local grid search and hardware limitations like GPU RAM. We do not claim that the model is entirely optimized, but optimized enough to perform well in our experiments and demonstrate the value and functionality of the method.

**Data:** Stocks we consider: AAPL, MSFT, IBM, AMZN and PG. These were chosen because they are large-cap, US stocks frequently traded on the NYSE. With the scripts available at [GitHub repository redacted for review] it is easy to download and process additional tickers if desired.

**Training:** To account for slight class imbalances in the dataset, we pass label weights proportional to the 1/(label frequency) to the Cross Entropy Loss calculation. As mentioned in Section 3, we train for 300 epochs on the synthetic data. However, on the real data (with far fewer unique structures than the entirely random $\gamma = 1$ synthetic dataset) the model converged after 80 epochs.

To train for 80 epochs on our hardware (see below) took 20 minutes and utilized 80% of the GPU, or $\approx 220$W.

**Model:** The model used in our experiments is modestly sized with 3 million trainable parameters. This could be made more compact, but we found that this configuration worked well in our setting. Figure 4 details the layers of the CNN representation learner and a single MLP head. $R_\theta$ outputs a vector of length 512 that is ingested by each of the MLP heads. In each MLP, Dropout1 = 0.5 and Dropout2 = 0.2, and it yields class conditional probabilities over each of the $k$ unique structure classes present in the dataset.

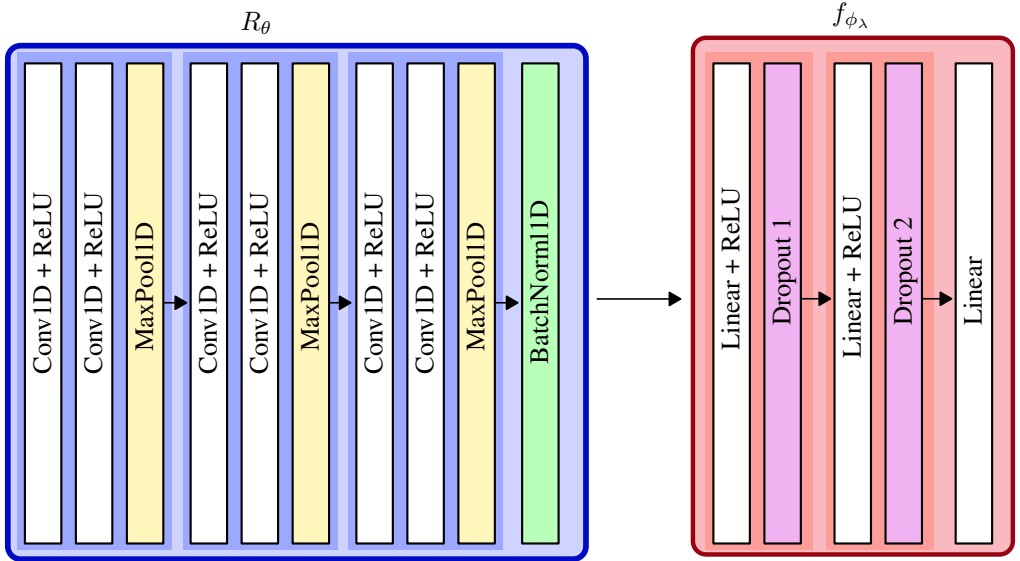

Figure 4: The internal structure of the CNN representation learner as well as the internal structure of a single MLP head (all MLPs are identical). For a complete definition of the model including all input and output sizes of each layer, see [GitHub repository redacted for review].

## A.1.1   HARDWARE

All training was performed on an internal cluster and utilized only a single GPU. Hardware specifics are as follows:

- Intel(R) Xeon(R) Silver 4214 CPU @ 2.20GHz

- NVIDIA GeForce RTX 2080 Ti
  - Driver Version: 460.73.01
  - Cuda Version: 11.2

### A.1.2 SOFTWARE

All training and model development was done using the latest stable PyTorch release. Complete setup scripts for Linux and Mac can be found at [GitHub repository redacted for review].

- Python 3.8.8
- PyTorch 1.8.1
- Scipy 1.6.2
- OpenCV 4.5.1.48 (used for resizing the input windows)
  - Interpolation: Cubic

### A.1.3 HYPERPARAMETERS

Training hyperparameters are as follows:

- Epochs: 300 for synthetic data, real-world data converged after 80 epochs.
- Batch Size: 128
- Optimizer: Adam
- Learning Rate: 0.001
- Weight Decay: 0
- Amsgrad: False
- Criterion: Weighted Cross Entropy Loss

### A.1.4 STANDARD ERRORS

Performance was not appreciably affected by random initialization conditions. As this these errors are calculated over four experiments, they would likely come down with additional trials. For clarity, they are expressed in terms of percent and rounded to the nearest whole number.

Standard Errors produced on synthetic data $X_{Synth}$:

| | Task-0 | | | Task-1 | | | Task-2 | | | Task-3 | | | Task-4 | | |
|---|---|---|---|---|---|---|---|---|---|---|---|---|---|---|---|
| Top-N Acc | 5 | 2 | 1 | 5 | 2 | 1 | 5 | 2 | 1 | 5 | 2 | 1 | 5 | 2 | 1 |
| $\gamma = 0$ | ±0 | ±0 | ±0 | ±0 | ±0 | ±0 | ±0 | ±0 | ±0 | ±0 | ±0 | ±0 | ±0 | ±0 | **±0** |
| $\gamma = 0.5$ | ±1 | ±1 | ±1 | ±1 | ±1 | ±1 | ±1 | ±0 | ±0 | ±1 | ±0 | ±0 | ±1 | ±0 | **±0** |
| $\gamma = 1$ | ±1 | ±1 | ±1 | ±2 | ±1 | ±1 | ±3 | ±2 | ±0 | ±0 | ±0 | ±0 | ±1 | ±1 | **±0** |

Table 3: Standard Error for each Task-$\lambda$ on synthetic datasets generated with varying amounts of randomness $\gamma$ in the structure sequences. All values are in percent and rounded to the nearest whole number.

Standard Errors produced on real-world data $X_R$:

| | Task-0 | | | Task-1 | | | Task-2 | | | Task-3 | | | Task-4 | | |
|---|---|---|---|---|---|---|---|---|---|---|---|---|---|---|---|
| | 5 | 2 | 1 | 5 | 2 | 1 | 5 | 2 | 1 | 5 | 2 | 1 | 5 | 2 | 1 |
| Top-N Acc | ±0 | ±0 | ±0 | ±1 | ±0 | ±0 | ±1 | ±0 | ±0 | ±2 | ±0 | ±0 | ±3 | ±1 | ±1 |

Table 4: Standard Error for each Task-$\lambda$ on the real-world price-action data. All values are in percent and rounded to the nearest whole number.

