# OpenReview forum: "Learning Predictive, Online Approximations of Explanatory, Offline Algorithms"
_ICLR.cc/2022/Conference — ICLR 2022 Submitted_

### Official Review · Reviewer_aaFn · 2021-11-02

**Correctness:** 4
**Technical Novelty And Significance:** 3
**Empirical Novelty And Significance:** 3
**Recommendation:** 6
**Confidence:** 2

**Main Review:**

In general, I like the idea of approximating the behavior of offline algorithms through the lens of multiple progressively-forward looking tasks (which essentially predicts the trajectory of future actions of the offline algorithm), since this allows us to predict further into the future (as opposed to predicting one-step ahead in standard ML methodologies). To the best of my knowledge, the idea of encoding the behavior of offline algorithms in graph structures, and then predicting the occurrence of such structures for multiple actions ahead via a multi-task learning framework is novel.

The following are some questions/concerns:
1. From my understanding, the ultimate goal of the whole paper is to approximate the behavior of offline algorithms in real time, as opposed to directly predicting the ground truth evolution of the time series. This seems to me that the proposed framework’s performance is primarily driven by how well the offline algorithm can fit the historical data. That being said, if the offline algorithm significantly overfits the offline data (e.g. some complex deep neural network), does this mean the offline-to-online framework can also perform arbitrarily well under certain conditions? If so, I find this hard to believe. I might be misunderstanding something here, and it would be great if the authors can provide some more explanations and insights in the paper (e.g. what are some key drivers for the proposed framework’s performance, and how does the proposed framework’s performance relate to that of the offline algorithm).

2. From a practical perspective, it seems to me that the algorithm is very "data hungry" as the number of structures may grow exponentially in the number of decision points in each structure. Hence, I believe there is this inherent trade-off between the amount of data required for labeling structures and how far we can predict into the future. The paper seems to be lacking detailed discussions for this tradeoff, or, on a related note, for how one should choose the "optimal" number of decisions within a structure.

3. I am confused about the occurrence moments of predicted future actions (since the proposed algorithm is predicting X actions ahead, instead of X moment ahead). Consider the stock market example, where we have task 1 that predicts 1 action ahead of some offline algo, and task 2 that predicts 2 actions ahead. How do we know that the last predicted action in task 2 is further away in the future than the (single) prediction action in task 1? In other words, from my understanding the predicted structures are completely agnostic to actual occurrence moments, and hence we cannot compare prediction actions across tasks? I might have missed related discussions in the paper, and it would be great if the authors can add some more emphasis.

4. I find the discussions in Section 2 General Schema quite difficult to digest at first read, and not until I went through the entire paper did I better understand how the multi-task learning framework works. Perhaps instead of discussing pure concepts (e.g. structure, actions, etc.), introducing the methodological framework within the context of a simple concrete example (e.g. a simplified version of the stock market example with some dummy offline algorithm) would improve the overall clarity of this section.


**Summary Of The Paper:**

This work studies a methodological framework to transform/approximate offline algorithms into their online counterparts. The main methodology is to predict an offline algorithm’s actions in the real time future via learning behavioral structures of the offline algorithm using past data. The work presents several experiments using both synthetic and real (stock market) data.

**Summary Of The Review:**

To the best of my knowledge, the proposed offline-to-online framework by predicting behavioral structures of the offline algo through a multi-task learning scheme is novel.

For weaknesses, more explanations/discussions on the following aspects would improve the paper: 1. How the performance of the proposed framework relates to that of the offline algorithm; 2. Choice for number of decisions in a structure; 3. Comparing predictions across different tasks. The paper’s exposition in terms of explaining the key concepts can also be improved.

---

> ### Author Response · Authors · 2021-11-20
> **Clarifying Points and Discussion**
>
> Thank you very much for your careful review. We address the concerns below:
>
> 1. For descriptive tasks, i.e. Task-0 where the entire structure we're trying to identify is visible in the window, a neural network should be able to perform arbitrarily well given sufficient training data. This is a similar to a standard classification problem. However, as we shift our focus into the future, i.e. Task-1 through Task-4 where 1/5 or 4/5 decision points are not present in the input window, the _predictive_ capacity of our method is determined by the degree to which there exist patterns in the sequences of behavioral structures present in the data. This is what our experiments on synthetic data were attempting to show. As we increase the randomness in the sequence of behavioral structures, we're only able to learn _up to_ that inherent randomness.
>
> 2. This is indeed a limitation of the current approach, which we address in item 2) of our response to Reviewer zoKR. In short, to handle the problem of a growing number of decision points yielding an exponentially growing number of classes, we could predict the graph structures directly instead of predicting the class label associated with a particular structure. And in regards to the optimal number of decision points to consider, that will be heavily application dependent.
>
> 3. The predicted structures are indeed agnostic to the temporal relationships between their constituent decision points. However, in the ideal case, as the tasks become increasingly forward looking, only those decision points in the future would change their order (because an ideal system would be able to extrapolate from the behavioral structure fully visible in the window). That said, adding a temporal component, i.e. predicting relative decision point positions and their spacing, could be a productive direction for future research.
>
> 4. In future revisions, we will add more concrete examples to build a more general intuition before getting into the finer details and notation (which has also been clarified after helpful suggestions from Reviewer gii5).
>
> Thank you again, and we hope that this response has clarified the questions put forth in your review.

---

### Official Review · Reviewer_zoKR · 2021-11-04

**Correctness:** 2
**Technical Novelty And Significance:** 3
**Empirical Novelty And Significance:** 3
**Recommendation:** 6
**Confidence:** 3

**Main Review:**

Strengths:


+ A novel formulation and research topic. The idea of trying to predict the behavior of an offline algorithm in a online setting using multi-task learning is a new approach. The exact formulation and the the way to pose this as MTL is non-trivial. The bulk of the contributions of this work is to make this modelling approach. Once figured out the proposed algorithm itself is standard multi-task learning.

+ This paper contributes to the now growing line of work on bridging classical algorithms with machine learning. In that line of work, this considered approach is novel. It gives a new perspective. The typical direction has been to use the ML model as hints to improve the online/offline algorithm. On the other hand here, the online to offline algorithm is bridged via a machine learning task.

+ For the most part the paper is clear and well-written.

Weakness

- The first main weakness I find in this paper is that, it does not sufficiently motivate the problem well. In particular, the online problem and it being posed as a MTL seems very abstract to the reader. It is not clear, how to use the outcome of this modelling in an actionable form. In particular, how does one interpret the class prediction for a window? What happens if the number of classes are unknown/evolving? May be elaborating this on a toy/standard offline algorithm before making it abstract would help the reader a great bit.

- Related to above, the formulation makes it seem like this applies to any offline algorithm. But it really only applies to offline algorithms that work on time-segmented data. So it comes of as over-selling the main contributions of the paper. Please correct me if I am wrong; if not, I would reword the introduction to make this aspect very clear.

**Summary Of The Paper:**

This paper considers the problem of an offline algorithm that operates on a time-series X to obtain sequence of decisions in an online setting. That is, it tries to approximate the behavior of this offline algorithm in a setting where at time t the algorithm only has access to the input until t (whereas in the offline algorithm the algorithm can lookahead and optimize). The pose this as a multi-task learning problem, where they slice the input into windows of size d, and the goal is to map each d dimensional window to one of the k possible structures in the dataset. They propose a MTL algorithm and use simulations and real-world stock market data to study the effects of their approach.

**Summary Of The Review:**

I like some of the ideas of this paper, but overall I think that it falls just below the bar because of the reasons I stated in the weakness. Please correct me if my understadning is incorrect.

---

> ### Author Response · Authors · 2021-11-20
> **Clarifying Points and Discussion**
>
> Thank you very much for your careful review. We address the concerns below:
>
> 1. Classes are associated with a particular graph-structure, so, following our example, interpretation involves decomposing the predicted graph into a time series. We agree that this could be clarified, and in future revisions we will include more description and a diagram to avoid confusion.
>
> 2. As it stands, the multi-task learning model yields probability distributions over a finite set of patterns already identified in the training set. We agree that this is a limitation. If instead of predicting pre-defined structures, we generated the graphical decision-point representations for those structures directly (i.e. our model generated the actual graph instead of predicting the class label associated with that graph), we could avoid this problem. This could also solve the problem of structures in the test set not being present in the training set, which could feasibly happen if the model is deployed in real-time on unseen data. This type of generative procedure fits within the scope of the proposed methodology, and we note that different applications will likely require different architectures for mapping raw input data to behavioral structures. That said, in future revisions, we will clarify this limitation.
>
> 3. This formulation does apply to arbitrary offline algorithms and we agree that the demonstration on time-series data, while intended to clarify the methodology, may lead to confusion. In future revisions, we will add additional experiments on problems outside the domain of time-series.
>
> Thank you again, and we hope that this response has clarified the questions put forth in your review.

---

### Official Review · Reviewer_gii5 · 2021-11-08

**Correctness:** 1
**Technical Novelty And Significance:** 3
**Empirical Novelty And Significance:** Not applicable
**Recommendation:** 3
**Confidence:** 4

**Main Review:**

**Novelty and significance.**

I am not an expert in this domain but to my knowledge the proposed approach is novel and presents an interesting method for using offline algorithms to create datasets for training machine learning models to approximate the outputs of the offline algorithms. I think the idea could be of interest to the community.

That said, the paper does not provide any way to evaluate the significance of the proposed result, as there are no empirical (or theoretical) comparisons to any other methods. Thus, it is impossible to situate the proposed method either relatively or absolutely to determine whether the method will be of any benefit to the community. The paper presents two datasets (a synthetic toy dataset and a constructed dataset of historical stock market data), neither of which seem to have been used in the literature before, and trains the proposed method on these datasets but compares to no other methods. The results show that the method has higher accuracy for the easier classification tasks and lower accuracy for the lower harder prediction tasks, and that the method seems to get above chance accuracy on most problems, but this does not tell the reader anything about the overall performance and behavior of the algorithm.

In future revisions of the paper, the authors should compare to other algorithms in this same space. A reasonable place to start is with the works discussed in the related work section. You can show generality by taking the proposed algorithm and comparing it with multiple different existing approaches on the different tasks that each of those existing approaches works on. If the scores of the proposed approach are reasonable, then we will have some evidence that it works as claimed. I urge the authors to also perform ablations on the method. What effects do changes in model architecture have? Or how does the choice of offline algorithm affect the method? How do variations of the synthetic dataset affect the proposed approach as opposed to other methods (i.e., is it more robust or more accurate in particular regimes, such as different values of n, |S|, \gamma, and d)? Note that two of the three proposed values of $\gamma = {0, 0.5, 1}$ are trivial and thus do not provide much information. I encourage the authors to also include $\gamma=0.25$ and $0.75$ to better show trends, and to plot these values instead of just putting the numbers in a table. Further, showing top-$k$ for $k=5,2,1$ seems unnecessary. 5 and 1 would be sufficient.

Regarding the claim of meeting or exceeding performance on ML-based stock prediction systems, there is no evidence given in this paper for this claim so it is unsubstantiated. As I understand it, the cited paper (Rezaei et al., 2021) uses an entirely different dataset, so comparisons of accuracy are meaningless.

**Clarity.**
Overall the method is fairly clearly explained, and the remainder of the paper is clear. I think the paper would benefit from providing a summary of the method at the beginning of section 2, and from some changes to notation to simplify the presentation and to fix some issues with the notation. The precise method to generate the synthetic dataset and create the stock market data should be detailed in the paper as well, without requiring readers to go to the (not yet provided) code.

**Detailed questions and comments.**
- Preprocessing both the train and test splits together is wrong as it allows information to bleed from test to train, both in the form of the normalization and the set of structures trained on. All pre-processing should be performed only on the train data, the statistics retained, and then these used on the test data.
- The fact that the number of unique structures $|S|$ changes for different values of $\gamma$ makes it difficult to compare trends across values of \gamma. Instead, I would suggest the authors change the dataset generation process to first specify an alphabet of structures $S$ and then generate (noised) trajectories from this alphabet.
- It appears that $\lambda$ is used both as an index and a count, but the count value of $\lambda$ always equals $n$, so why not just use $n$?
- Defining $|S| = k$ is confusing as $k$ is already (and typically) used as an index variable and it is nonstandard and unclear to use it as a count.
- Please define a domain for the class labels and use that directly to simplify notation.
- The definition of a window seems to assume that decision points are uniformly spaced, but this is not made explicit anywhere.
- The definition of the estimator $f$ in eq (2) does not match the text, as it should be mapping onto the simplex of the class label domain based on the corresponding text.
- Please explain the method for computing the decision points (l1TF) in more detail.


**Summary Of The Paper:**

The paper presents a novel method for approximating offline time-series algorithms in an online setting. The method achieves this by assigning each window of the time-series data to a set of discrete classes based on the behavioral structure in that window, where the behavioral structures are encodings of the relative placements of the decision points in that window as determined by an offline time-series algorithm. These classes then provide the targets for a series of connected classification problems. An approximate online algorithm is obtained by training a multi-task classification neural network to solve these. Results on one-dimensional synthetic and stock-market data show that the predictive behavior of this method matches our intuitions, where it is most accurate when explaining the data and least accurate when predicting into the future.

**Summary Of The Review:**

Overall, this paper lacks an evaluation for the proposed method, and thus cannot be accepted. The proposed approach seems interesting, and I encourage the authors to resubmit after incorporating a proper evaluation by comparing to other methods on established datasets and addressing some of the other comments above (in particular the dataset issues).

---

> ### Author Response · Authors · 2021-11-20
> **Clarifying Points and Discussion**
>
> Thank you very much for your careful review. We address the concerns below:
>
> 1. While a direct comparison to existing methods is difficult - because ours is the first to approach the problem in the general case - this critique is common across reviews and we take it seriously. In future revisions we will ensure to properly differentiate our method from the existing literature and further clarify its performance by comparing our results against the domain-specific methods presented in the Related Work section.
>
> 2. In regards to ablation studies, between a variety of multi-task neural network architectures we did not observe meaningful differences in predictive performance - neither from the initialization (which we mention in footnote 6) nor the architecture (so long as it was a multi-task architecture). In regards to ablation studies, we tried a number of novel architectures using soft and hard parameter sharing schemes having symmetric and asymmetric links between task heads. The architecture used in the paper was essentially the result of repeated ablation studies, and is the simplest multi-task architecture that performed well on our experimental data. We note that multi-task architectures are well suited to this problem and did outperform other more naive architectures like a simple MLP.
>
> 3. Thank you for calling our attention to notational confusion around $\lambda$ and $|S|$, this has been clarified.
>
> 4. We will go further into detail about the L1TF algorithm in future revisions.
>
> 5. In future revisions, we will compare our method against other domain-specific approaches and use identical input data so comparisons are clear and fair.
>
> Thank you again, and we hope that this response has clarified the questions put forth in your review.

---

> > ### Comment · Reviewer_gii5 · 2021-11-25
> > **Response to rebuttal**
> >
> > Thank you for your response. As there has been nothing added, my score will not change. However, I again strongly encourage the authors to make the changes they've outlined above, as that is necessary for considering acceptance of the paper.

---

### Official Review · Reviewer_EgW9 · 2021-11-08

**Correctness:** 1
**Technical Novelty And Significance:** 3
**Empirical Novelty And Significance:** 2
**Recommendation:** 3
**Confidence:** 4

**Main Review:**

**Strengths**
+ The paper describes their method, experimental setup, and results very clearly.
+ The paper presents an interesting research direction, using knowledge from offline algorithms to improve performance of online algorithms via learning.
+ The paper highlights that leveraging these methods could be impactful for many domains.

---
**Weaknesses**
+ The primary weakness is the main claim seem incorrect.
The authors claim to develop a general framework for **approximating offline algorithms using online algorithms**. But the online algorithms trained in this paper do not directly attempt to approximate the offline algorithms. The online algorithms do not even produce the same type of outputs as the offline algorithms.

The offline algorithms take time-series X={x_1, ..., x_T} as input and produces outputs in the form of decision points A(X) = {(x_i, a_{x_i}), ... (x_j, a_{x_j})}.

An online algorithm that approximates this could take as input a partial time-series X_t = {x_t-d+1, ..., x_t} and decide whether or not to produce a decision point at time t.

Instead the online algorithms in this work predict class labels which are a lossy mapping of sequences of decision points. As a result it is not clear to me in what sense these online algorithms approximate the offline algorithms.

Can you clarify this for me? In what sense is it approximating the offline algorithm? If classification accuracy is 100%, can we make any statements about how good the approximation is?

+ Because the primary claim is not clear, it's not clear how to evaluate the proposed method, or what baselines to compare to.
+ The related work section is short and only mentions offline to online conversion, and explanatory vs predictive models. Since the paper also mentions time-series forecasting, it would be good mention related work in that field too.

**Suggestion**
+ May I suggest the following claim: the paper develops a method which leverages offline algorithms to perform better online time-series prediction.
+ Then the main evaluation metric should be time-series prediction. And baselines would include a range of methods for time-series prediction, and ablations which use offline outputs in different ways but have similar architecture.
+ It would be good to report the performance of comparable ML time-series prediction algorithms trained on the same data, and with similar architectures. Currently the authors mention another paper but do not report numbers for it.
+ Related: In Introduction paragraph 2, you compare your method to time-series forecasting techniques. And mention 3 benefits of your technique which focuses on behavior, vs techniques that directly predict time-series trajectories. It would be good to see this demonstrated experimentally.

**Summary Of The Paper:**

This paper makes use of **offline algorithms** (i.e. algorithms that can view entire time-series) to produce outputs which are used to train an **online algorithm** (i.e. an algorithm that can only view past values of a time-series). The online algorithms are not trained to produce the outputs of the offline algorithm directly, instead windows of the outputs are mapped to class labels using a hand-crafted mapping specific to the domain. The online algorithms are then trained to predict the class labels of the current window and progressively forward-looking windows (i.e. a multi-task prediction problem) given a window of the time-series.

They apply this method to synthetic and real-world time-series data (historical stock market data), and report the classification accurately of each of the multi-task prediction problems.

They also mention this can be used to predict the direction the price of a stock will move. And state that their method is competitive with state-of-the-art ML methods on this task.


**Summary Of The Review:**

While the ideas presented in this work could be very impactful, as the paper is currently written, its main claim seems incorrect which is grounds for rejection.

The paper claims to develop a general framework for approximating offline algorithms using online algorithms. But to me, it seems the online algorithms do not approximate the offline algorithms.

I think the paper could be made substantially better in one of two ways:
1) The authors clarify in what sense the online algorithms approximate the offline algorithms.
2) The authors modify the claims to more accurately reflect the the method, and add additional experiments to support those claims.

---

> ### Author Response · Authors · 2021-11-20
> **Clarifying Points and Discussion**
>
> Thank you very much for your careful review. We address the concerns below:
>
> 1. Our approach is approximating offline behavior in real-time by using predicted future behavioral patterns to inform our understanding in the present. However, instead of forecasting the underlying time series directly, we are forecasting the behavior of the offline algorithm on those data. This concept will be developed and explained more fully in future revisions.
>
> 2. We agree that clarifying the method's relation to existing time series forecasting techniques would help to differentiate it. In future revisions, we will compare our method (predicting decision points instead of trajectories) to existing time series forecasting techniques (predicting trajectories directly).
>
> While we agree that this method does leverage offline algorithms to perform better in time series forecasting tasks, we believe the method is more general than that and will take extra care to clarify what exactly our method is approximating.
>
> Thank you again, and we hope that this response has clarified the questions put forth in your review.

---

### Official Review · Reviewer_s1H9 · 2021-11-11

**Correctness:** 3
**Technical Novelty And Significance:** 4
**Empirical Novelty And Significance:** 4
**Recommendation:** 5
**Confidence:** 2

**Main Review:**

*Strengths*:
  1.  The motivation of bridging the gap between offline algorithms and their online counterparts is clear and practical. Real-world examples are discussed in the introduction and conclusion, and help to further understand the motivation.
  2.  The proposed approach is novel to my knowledge. I admire the idea to capture the behavior structure by multi-task learning model, which is interesting to create datasets using offline algorithm for training the online counterpart.
  3.  The design is clearly presented. Figure 1, 2 are helpful to understand the high-level framework.

*Weakness*:
  1.  Why no baselines are presented in the experiment part. I am not an expert in this field, so I am not entirely convinced that it needs any comparison of other benchmarks.
  2.  Is there any theoretical guarantees or insights behind the design?
  3.  I personally think that the paper writing can be further enhanced. For example: 1) The sections and subsections does not follow a traditional manner, e.g., the experiment and experimental results are not in one section; the ethics is a subsection of conclusion. 2) Although the authors claim that the proposed method outperform the SOTA, however, the performance of the SOTA model is not present in the Table.

Minor:
``We review this limitation more thoroughly in Section ??’’ in page 6 —> Section ??

**Summary Of The Paper:**

This work claims to propose a general methodology for approximating offline algorithms in online settings, in contrast to previous methods only for particular cases. To achieve this, the author prosed a multi-tasks-based method to learn from the datasets created by the offline algorithms. Experiments are conducted to verify the idea.

**Summary Of The Review:**

I admire the motivation, idea, and possible impact of this paper. However, I am not entirely convinced that the experimental results are convincing enough. I would like to update the score after interacting with the authors and other reviewers.

---

> ### Author Response · Authors · 2021-11-20
> **Clarifying Points and Discussion**
>
> Thank you very much for your careful review. We address the concerns below:
>
> 1. One-to-one comparison is difficult in our case as the general problem of approximating arbitrary offline algorithms has not been directly addressed in the literature.
>
> 2. We do not provide theoretical guarantees for our method in this paper because part of the advantage of the proposed methodology is that its ultimate form is a straightforward mapping between source and target domains. As such, its theoretical guarantees are the same as those of existing ML algorithms and subject to the same challenges of optimization and stability.
>
> Thank you again, and we hope that this response has clarified the questions put forth in your review.

---

### Decision · Program_Chairs · 2022-01-20

**Decision:**

Reject

**Comment:**

## A Brief Summary
This paper uses offline algorithms that can see the entire time-series to approximate the online algorithms that can only view the past time-series. The way this is done is basically, the offline algorithm is used to provide discrete class targets to train the online algorithm. The paper presents results on synthetic and historical stock market data.

## Reviewer s1H9
**Strengths:**
- Practical problem.
- Novel approach.
- Clear presentation.
**Weaknesses:**
- No other baselines.
- No theoretical guarantees behind the approach.
- Writing could be improved.

## Reviewer EgW9
**Strengths:**
- Clear writing.
- Interesting research direction.
**Weaknesses:**
- The primary claim seems incorrect and unclear.
- Due to the unclarity about the primary claim of this paper, it is difficult to evaluate the paper.
- Lack of baselines.
- The lack of discussions of the related works.

## Reviewer gii5
**Strengths:**
- Interesting and novel approach.
**Weaknesses:**
- Difficult to evaluate, with no empirical baselines or theoretical evidence.
- The datasets used in the paper are not used in the literature before. Authors should provide experimental results on datasets from the literature as well.
- The paper needs to compare against the other baselines discussed in the related works.
- More ablations and analysis on the proposed algorithm is required.
- Unsubstantiated claims regarding being SOTA on the task, since the paper doesn't compare against any other baselines on these datasets.
- The paper can be restructured to improve the flow and clarity.

## Reviewer zoKR
**Strengths:**
- Novel and interesting research topic.
- Bridging classical algorithms and ML.
- Clearly written.

**Weaknesses:**
- Lack of motivation for the problem.
- The approach only works with offline algorithms that work on time-segmented data.

## Reviewer aaFn
**Strengths:**
- Novel algorithm.

**Weaknesses:**
- Potentially overfitting to the offline data.
- Data hungry approach.
- Confusion related to the occurrence moments of predicted future actions.
- Section 2 is difficult to understand.

## Key Takeaways and Thoughts
Overall, I think the problem setup is very interesting. However, as pointed out by reviewers gii5 and EgW5, due to the lack of baselines, it is tough to compare the proposed algorithm against other approaches, and this paper's evaluation is challenging. I would recommend the authors include more ablations in the future version of the paper and baselines and address the other issues pointed out above by the reviewers.